# Parent-Adolescent Communication, School Engagement, and Internet Addiction among Chinese Adolescents: The Moderating Effect of Rejection Sensitivity

**DOI:** 10.3390/ijerph18073542

**Published:** 2021-03-29

**Authors:** Jingjing Li, Chengfu Yu, Shuangju Zhen, Wei Zhang

**Affiliations:** 1Center for Studies of Psychological Application, School of Psychology, South China Normal University, Guangzhou 510631, China; 2020010222@m.scnu.edu.cn (J.L.); shuangjuzhen@foxmail.com (S.Z.); 2Department of Psychology, Research Center of Adolescent Psychology and Behavior, School of Education, Guangzhou University, Guangzhou 510006, China; yuchengfu@gzhu.edu.cn

**Keywords:** parent-adolescent communication, Internet addiction, school engagement, rejection sensitivity

## Abstract

Although a large body of research has indicated that parent-adolescent communication is a crucial protective factor for adolescent Internet addiction, the mediating and moderating mechanisms underlying this relationship remain unclear. To address this research gap, this study, based on social control theory and the organism-environment interaction model, was designed to test whether school engagement mediated the relationship between parent-adolescent communication and adolescent Internet addiction and whether this mediating effect was moderated by rejection sensitivity. A sample of 1006 adolescents (*Mean_age_* = 13.16 years, *SD* = 0.67) anonymously completed the questionnaires. The results showed that the positive association between parent-adolescent communication and adolescent Internet addiction was mediated by school engagement. Moreover, this indirect link was stronger among adolescents with high rejection sensitivity than those with low rejection sensitivity. These findings highlighted school engagement as a potential mechanism linking parent-adolescent communication to adolescent Internet addiction, with high rejection sensitivity being an important risk factor amplifying this indirect effect. Intervention programs aimed at reducing Internet addiction among adolescents might benefit from the current research.

## 1. Introduction

With the rapid development of the network, the Internet has become an important platform for exchanging ideas, leisure, acquiring knowledge, and entertainment. However, uncontrolled and excessive Internet use causes “Internet addiction,” which is widely used to describe a phenomenon wherein persons are unable to control their Internet use [1]. According to survey research, approximately 8% of adolescents in China suffer from Internet addiction [2,3,4]. Internet addiction is related to various premorbid symptoms and clinical disorders [4], which threaten academic performance, sleep quality, mental health and emotional adaptation. Furthermore, it is becoming a main global public health issue that has attracted the attention of researchers from numerous fields [5,6,7]. Therefore, to establish an effective control measure, it is necessary to identify risk factors or underlying mechanisms that might make individuals more prone to Internet addiction.

Inherent parent- and family-related factors play a crucial role in the prevalence of Internet addiction [4]. Parent-adolescent communication (PAC) is an important aspect of the family system and thereby, is closely related to Internet addiction. Based on family systems theory [8,9], parent-adolescent communication is beneficial for an individual’s development because a positive parent-child relationship confers a feeling of being loved and cared for, which fulfills an individual’s needs for belonging and affection [4]. Individuals with positive parent-adolescent communication are less likely to be addicted to the Internet compared with those who have negative parent-adolescent communication with their parents [8]. According to this perspective, many studies have shown that negative parent-adolescent communication will increase the possibility of Internet use and eventually result in Internet addiction [9].

Recently, the mediating and moderating mechanisms underlying the relationship between parent-adolescent communication and Internet addiction have begun to be examined [5,9]; for example, studies proved that deviant peer affiliation [3,4] and self-esteem [5] are potential mediators. According to the organism-environment interaction model [6,7,8,9], the individual and environment constitute a complex system and all factors in the system do not act independently but rely on each other and work together. School is the most important place for student development. School engagement might function as a protective factor that prevents Internet addiction and promotes a healthy life. Researchers have found that individuals who are attached to school, are behaviorally engaged, or participate in school activities are less likely to be addicted to the Internet [3,4]. Therefore, based on the organism-environment interaction model and family systems theory, this study was designed to test whether school engagement mediates the relationship between parent-adolescent communication and adolescent Internet addiction and determine whether this mediating effect is moderated by sensation seeking. This was performed to more systematically reveal the delaying mechanism of adolescent Internet addiction and provide a basis for its scientific prevention and effective control.

### 1.1. Parent-Adolescent Communication and Adolescent Internet Addiction

Inherent parent- and family-related factors play a significant role in preventing Internet addiction [4,8,9]. The family’s influence on adolescent Internet addiction has always been an important issue, especially the relationship between adolescent Internet addiction and family from the perspective of parent-adolescent communication, which has been of interest to several researchers [10,11]. Parent-adolescent communication is a crucial aspect of the family system and refers to the adolescents’ perception of communication with their parents [12]. Positive parent-adolescent communication can help reduce adolescents’ Internet addiction [13,14,15], whereas negative parent-adolescent communication will increase the possibility of Internet addiction among teenagers [16,17,18]. Park et al. [5] found that parent-child communication quality in high school students with Internet addiction was significantly lower than that of the non-addiction group. Moreover, Lam [16] performed an intervention for adolescent Internet addiction to promote parent-adolescent communication and achieved good therapeutic results.

According to family systems theory [19], the structure and organization of a family are important factors that strongly predict and influence adolescent behaviors. Adolescents with negative parent-adolescent communication generally engage in Internet addiction to satisfy their unmet family needs [20]. Empirical evidence showed that poor parent-adolescent communication positively predicted adolescent Internet addiction [21,22] because adolescence who lack family warmth will meet their psychological needs in other ways and Internet is the most convenient way for adolescents to meet their psychological needs [23]. Among others, online gaming upgrades can meet their competence needs, making friends can meet their relatedness needs, and immersing and laying freely in the online world can meet their autonomy needs. Parent-adolescent communication is helpful to meet children’s psychological needs, which in turn helps prevent teenagers from meeting their psychological needs through other means such as Internet addiction [16,18,23].

### 1.2. School Engagement as a Potential Mediator

The question remains of how parent-adolescent communication affects adolescent Internet addiction. According to social control theory [24], environmental factors (such as parent-adolescent communication) affect individuals’ behavior by strengthening or weakening the degree of connection between individuals and social organizations (such as school engagement). School is one of the most important places for adolescent development [25]. School engagement plays a protective role in preventing Internet addiction and promotes healthy lifestyles [26]. It is also a multifaceted concept including cognitive, behavioral, and emotional engagement, which represent the degree of students’ relationships with the school, as well as their participation in school activities [27]. Researchers found that adolescents who are emotionally attached to school, are behaviorally engaged, or participate in school activities are less likely to be addicted to the Internet because such adolescents are more likely to develop positive relationships with teachers and their peers and generally strive to meet society’s expectations [23]. Research has also shown that individuals who experienced positive pathways of school engagement were unlikely to be involved in Internet addiction [23,28,29]. This is because school engagement limits the energy and time available for adverse activities (such as, smoking, alcohol abuse etc.) and strengthens individuals’ connections to their institutions [23,30].

Parent-adolescent communication promotes adolescents’ school engagement [23,31,32]. First, school engagement is optimized and responsive to contextual characteristics when adolescents perceive a social context to support their developmental needs [33,34]. The need for relatedness is one of these developmental needs and is likely to arise when parents create a harmonious family environment for adolescents [23,32]. Strong relationships between adolescents and their parents (such as parent-adolescent communication) can function as a motivational resource when adolescents are confronted with challenges and difficulties at school [23,35]. In addition, growing evidence associates parent-adolescent communication to higher school engagement [23,35,36]. Second, studies have confirmed that family intimacy and parenting styles can significantly predict cognitive engagement [37]. Adolescents form self-regulated strategies in the process of parent-adolescent communication, solidifying these self-regulated strategies in their own internal working mode, and thus, these have a far-reaching impact on their behavior in the future [38]. Moreover, to a large extent, the beliefs of self-regulated strategies come from parents’ education [39]. Emotional warmth in family parenting styles can positively predict adolescents’ self-regulated strategies [40], whereas teenagers’ self-regulated ability will affect Internet use and other delinquency behaviors. Therefore, school engagement is an important pathway for parent-adolescent communication to influence adolescents’ Internet addiction. Hence, the current research proposes the following assumption:

**Hypothesis** **1** **(H1).**
*School engagement mediates the association between parent-adolescent communication and adolescent Internet addiction.*


### 1.3. Rejection Sensitivity as a Moderator

Although parent-adolescent communication was generally believed to be a protective index for promoting school engagement [23] and reducing Internet addiction [16,17,18], it might not bring about a positive effect for all adolescents. Based on the organism-environment interaction theory [41], adolescent Internet addiction is determined by the interaction effect between environmental and individual factors (e.g., rejection sensitivity). Rejection sensitivity refers to individuals’ tendency to anxiously anticipate negative information and easily perceive rejection information, as well as the degree to which individuals tend to overreact [42,43]. Rejection sensitivity is a crucial risk factor for adolescent Internet addiction [44,45,46,47]. This is mainly because the Internet provides a place to escape the frustration of interpersonal rejection [48]. Furthermore, in recent years, a growing body of research has paid attention to the joint contribution of environmental and individual factors to Internet addiction [49,50,51]. Personal and contextual factors are not incompatible in the organism-environment interaction theory [41]; thus, they should be regarded simultaneously to explain Internet addiction [52]. For example, the two types of control factors might interact with one another in explaining Internet addiction. Specifically, in this study, high rejection sensitivity might amplify the adverse effect of negative contextual factors (e.g., school) on adolescents’ problem behaviors (e.g., Internet addiction). Empirical studies have also shown that rejection sensitivity can significantly moderate the influence of school-related factors on adolescent development [53,54,55,56,57]. For example, Ylenia et al. [47] found that high rejection sensitivity could serve as a risk factor that can significantly influence adolescents’ low school academic self-concept on their externalizing behaviors. Similarly, Morrow et al. [57] found that the negative relationship between school violence and depression was stronger for adolescents with high rejection sensitivity than for those with low rejection sensitivity. Based on this empirical evidence and theoretical model, we propose the following assumption:

**Hypothesis** **2** **(H2).**
*Rejection sensitivity moderates the indirect relationship between parent-adolescent communication and adolescent Internet addiction via school engagement. Specifically, this indirect effect would become stronger for adolescents with high rejection sensitivity and weaker among those with low rejection sensitivity.*


In summary, the current study brings together two theories (the social control theory and the organism-environment interaction theory) to account for the mechanisms of how and when parent-adolescent communication is linked to adolescent Internet addiction. The first and second hypotheses, taken together, constitute a moderated mediation model shown in Figure 1.

## 2. Methods

### 2.1. Participants

Participants were 1006 adolescents (51.79% females, *n* = 521; 48.21% males, *n* = 485), recruited from three public junior middle schools in Guangdong Province in southern China. In this study, the stratified and random cluster sampling method was adopted to select participants. Participants in this study ranged from 12 to 15 years of age (average age: 13.16 years, *SD* = 0.67 years).

### 2.2. Measures

#### 2.2.1. Parent-Adolescent Communication

Parent-adolescent communication was assessed with the Chinese version of the Parent-Adolescent Communication Scale [58]. Participants were asked to report how often they communicate with their parents about academics, safety, interpersonal interaction, daily life, and emotional issues. A three-point Likert scale was used, with scores ranging from “1 = never,” 2 = “sometimes,” and “3 = often.” The average score was calculated, with a higher score indicating high-level parent-adolescent communication. In this study, the scale demonstrated excellent reliability (α = 0.91).

#### 2.2.2. Internet Addiction

Internet addiction was assessed using a 9-item scale adapted from the Internet Gaming Disorder Questionnaire [59]. Adolescents were asked to report how frequently they felt dependent on the Internet (e.g., “Have you deceived any of your family members, therapists, or others because of the amount of your Internet activity?”) on a 3-point Likert scale as follows: “1 = never,” “2 = sometimes,” and “3 = often.” The average score was calculated, with a higher score indicating a higher tendency toward Internet addiction. In this study, the scale demonstrated good reliability (*α* = 0.74).

#### 2.2.3. School Engagement

School engagement was assessed using the School Engagement Scale [32]. Adolescents were asked to report their behavioral, cognitive, and emotional engagement. Behavioral and cognitive engagement were rated from 1 = “never” to 5 = “always,” and emotional engagement was rated from 1 = “fully disagree” to 5 = “fully agree.” The average score was calculated, with a higher score indicating high-level school engagement. In this study, the scale demonstrated excellent reliability (*α* = 0.89).

#### 2.2.4. Rejection Sensitivity

Rejection sensitivity was measured using the Chinese version of the Rejection Sensitivity Scale [60]. This scale consists of 18 items assessing the feelings related to interpersonal experiences (e.g., “I’m very sensitive to rejection”). Items were rated on a 5-point Likert-type response scale ranging from 1 = “not at all true” to 5 = “always true.” The average score was calculated, with a higher score indicating a high-level rejection sensitivity. In this study, the scale demonstrated excellent reliability (α = 0.77).

#### 2.2.5. Control Variables

We included adolescents’ gender, age, and sensation-seeking as control variables because prior studies have documented that these variables were significantly associated with Internet addiction [26]. Sensation-seeking was assessed with the sensation-seeking subscale of the UPPS-P Impulsive Behavior Scale [61]. Adolescents rated how true each statement was (e.g., “I sometimes like doing things that are a bit frightening”) on a 4-point Likert scale ranging from 1 = “strongly disagree” to 4 = “strongly agree.” The average score was calculated, with a higher score indicating high-level sensation-seeking. In this study, the scale demonstrated excellent reliability (*α* = 0.74).

### 2.3. Procedure and Statistical Analyses

The survey materials and study procedures were approved by the Ethics in Human Research Committee of the Department of Psychology, Guangzhou University (protocol number: GZHU2019012; date of approval: 27 May 2019). The data were collected by well-trained graduate students majoring in psychology and professional psychology teachers. SPSS 25.0 (IBM, Armonk, NY, USA) was used for the descriptive statistical analysis. Moreover, mediation and moderation effects were tested with Mplus 7.2 [62]. Missing values were handled via full information maximum likelihood estimation, and bootstrapping analysis with 1000 replicates was performed to verify the significance of the paths. According to Hoyle’s suggestion [63], a model fit is considered acceptable when *χ*^2^/*df* is less than 5, CFI and TLI are greater than 0.90, and RMSEA and SRMR are less than 0.08.

## 3. Results

### 3.1. Preliminary Analyses

As shown in Table 1, the results indicated that parent-adolescent communication was positively correlated with school engagement and negatively correlated with Internet addiction. Second, school engagement was negatively correlated with Internet addiction. Moreover, rejection sensitivity was negatively correlated with school engagement and positively correlated with Internet addiction.

### 3.2. Testing for the Mediation Effect of School Engagement

The mediation model is represented in Figure 2; it had an excellent fit to the data as follows: *χ*^2^/*df* = 2.52, CFI = 0.96, TLI = 0.943, RMSEA = 0.039, and SRMR = 0.032. After controlling for gender, age, and sensation-seeking, it was found that parent-adolescent communication positively predicted school engagement (*β* = 0.40, *t* = 14.19, *p* < 0.001, 95% confidence interval [CI] [0.34, 0.45]), and school engagement negatively predicted Internet addiction (*β* = −0.30, *t* = −8.82, *p* < 0.001, 95% CI [−0.37, −0.23]). Moreover, the residual effect of parent-adolescent communication on Internet addiction was also significant (*β* = −0.07, *t* = −2.09, *p* < 0.05, 95% CI [−0.13, −0.003]). Bootstrapping analyses indicated that school engagement significantly mediated the relationship between parent-adolescent communication and adolescent Internet addiction (indirect effect = −0.12, *SE* = 0.02, 95% CI [−0.16, −0.09]).

### 3.3. Testing for the Moderated Mediation

The moderated mediation model is represented in Figure 3; it had a very good fit to the data as follows: *χ*^2^/*df* = 3.24, CFI = 0.935, TLI = 0.902, RMSEA = 0.047, and SRMR = 0.039. The bias-corrected percentile bootstrap results indicated that the indirect effect of parent-adolescent communication on adolescent Internet addiction through school engagement was moderated by rejection sensitivity. Specifically, rejection sensitivity moderated the association between school engagement and Internet addiction (*β* = −0.11, *t* = −3.23, *p* < 0.01, 95% CI [−0.18, −0.04]). We conducted a simple slopes test, and as depicted in Figure 4, the negative association between school engagement and Internet addiction was significantly stronger among adolescents with higher rejection sensitivity (1 *SD* above the mean; *β* = −0.39, *t* = −9.11, *p* < 0.001, 95% CI [−0.48, −0.31]) than among adolescents with lower rejection sensitivity (1 *SD* below the mean; *β* = −0.17, *t* = −4.16, *p* < 0.001, 95% CI [−0.25, −0.09]). Furthermore, parent-adolescent communication had a significant negative association with school engagement (*β* = 0.39, *t* = 13.57, *p* < 0.001, 95% CI [0.32, 0.45]), and rejection sensitivity had a significant negative association with school engagement (*β* = −0.08, *t* = −2.35, *p* < 0.05, 95% CI [−0.13, −0.01]) and a significant positive relationship with Internet addiction (*β* = 0.28, *t* = 8.48, *p* < 0.001, 95% CI [0.22, 0.34]). However, the interaction between adolescent communication and rejection sensitivity in predicting school engagement (*β* = −0.06, *t* = −1.86, *p* > 0.05, 95% CI [−0.12, 0.00]) and Internet addiction (*β* = −0.02, *t* = −0.70, *p* > 0.05, 95% CI [−0.08, 0.04]) was not significant.

Finally, the bias-corrected percentile bootstrap results indicated that the indirect link between parent-adolescent communication and Internet addiction via school engagement was stronger among adolescents with high rejection sensitivity (indirect effect = −0.13, *SE* = 0.03, 95% CI [−0.19, −0.09]) than among those with low rejection sensitivity (indirect effect = −0.08, *SE* = 0.02, 95% CI [−0.12, −0.04]). Therefore, the mediating effect of school engagement between parent-adolescent communication and adolescent Internet addiction was moderated by rejection sensitivity.

## 4. Discussion

This study examined how parent-adolescent communication relates to adolescent Internet addiction and whether the association varied according to their rejection sensitivity. This investigation found that adolescents with positive parent-adolescent communication showed more school engagement, which in turn relates to less Internet addiction. Furthermore, this indirect link is moderated by adolescents’ rejection sensitivity. The present finding enhances our understanding of how parent-adolescent communication relates to adolescent Internet addiction and contributes to identify underlying mechanism for adolescent Internet addiction.

In line with Hypothesis 1, this result showed that school engagement mediated the relationship between adolescent Internet addiction and parent-adolescent communication. According to the attachment theory [38], positive parent-adolescent communication enables adolescents to believe that they are loved and cared for and that the surrounding environment is safe and trustworthy, which encourages them to actively explore their surrounding environment and promotes school engagement. Adolescents with positive school engagement were more likely to interact with school authorities such as teachers, which is helpful for them to internalize the ruling principle. Furthermore, adolescents with higher school engagement were more likely to obey rules in school, which reduced the possibility of developing Internet addiction. Prior evidence has agreed with this view, showing a strong association between Internet addiction and school engagement [64,65]. These results correspond with the social control theory [24], which argues that people internalize the conduct rules and codes of social contexts as they have social connections with school institutions. Although previous studies have separately examined the influence of parent-adolescent communication or school engagement on adolescent Internet addiction [64,65], the present study examined those factors simultaneously and documented that school engagement should be a significant explanatory mechanism for why positive parent-adolescent communication reduces Internet addiction among adolescents. Our study showed that school engagement is a crucial psycho-social asset that can be cultivated by positive parent-adolescent communication. This has a significant contribution to the mechanisms preventing adolescent Internet addiction.

Considering Hypothesis 2, our findings are consistent with the organism-environment interaction model [41], which supposed that behavior is determined by both individual and environmental factors. Our results indicated that rejection sensitivity can significantly enhance the adverse effects of low school engagement on adolescent Internet addiction. First, for adolescents with low rejection sensitivity, the Internet addiction tendency is always lower than that for those with high rejection sensitivity. Second, the Internet addiction tendency of adolescents with high rejection sensitivity decreases with an increase in school engagement. This could be because adolescents with high rejection sensitivity were more negatively affected by low school engagement and were more likely to use the Internet to seek solace and/or escape from reality. In contrast, adolescents with low rejection sensitivity have good interpersonal relationships; thus, to some extent, these can relieve the adverse influence of low school engagement [28,29,56]. Therefore, the results of this research suggested that the contribution of school engagement and rejection sensitivity to Internet addiction should not be viewed independently but in tandem. This reminded researchers to address different factor roles influencing internet addiction simultaneously. These findings also emphasize the importance of moderated mediation models. Compared to a simple mediating model, this study provided specific information that such a mediating effect might not be applicable to every adolescent, thereby providing better guidance for Internet addiction intervention.

There are several limitations to the present research that future studies should address. First, although cross-sectional research provides useful information about the relationships among different variables, a complete autoregressive cross-lagged design would be more beneficial for examining the direction between the main variables in this study. Second, data collection in the present research was based on adolescents’ self-reports. Although adolescents are more aware of their own mental condition than their parents [15,65] and perceiving parent-adolescent communication has a crucial impact on their development [15,65], multiple information points (such as parent report, self-report, and teacher report) will be beneficial in providing a more rigorous test for this study’s hypotheses. Finally, the current study focused on general Internet addiction. Future research should further examine the influencing factors on addiction based on specific Internet activities (e.g., Internet gaming addiction).

Although the current study has some shortcomings, our results also have significant practical contributions. First, this study confirmed the value of functional parent-adolescent communication. Positive parent-adolescent communication might help adolescents develop school engagement to prevent them from being addicted to the Internet. Therefore, it can be a focus for future Internet addiction intervention programs. Second, school engagement helps reduce adolescent Internet addiction. School engagement is malleable, and previous intervention studies have demonstrated its significant function in helping individuals deal with stress and challenges [15]. Third, we found that rejection sensitivity not only directly increased adolescent Internet addiction but also weakened the salutary impact of parent-adolescent communication on school engagement and the influence of school engagement on adolescent Internet addiction. Therefore, resilience resources, such as rejection sensitivity, should be emphasized. Finally, the moderated mediation model in the current research showed that systematic, integrated programs that consider both individual and environmental factors simultaneously are needed to prevent adolescent Internet addiction.

## 5. Conclusions

In conclusion, by examining a moderated mediation model that includes both environmental (school) and individual (rejection sensitivity) factors simultaneously, the current study promotes our understanding of when and how positive parent-adolescent communication reduces adolescent Internet addiction. Overall, positive parent-adolescent communication can promote adolescents’ school engagement, which in turn reduces adolescent Internet addiction. Moreover, the conductive effect of school engagement is weakened when adolescents have high levels of rejection sensitivity. These novel findings emphasize the importance of jointly examining environmental and individual factors to better understand the etiology of adolescent Internet addiction.

## Figures and Tables

**Figure 1 ijerph-18-03542-f001:**
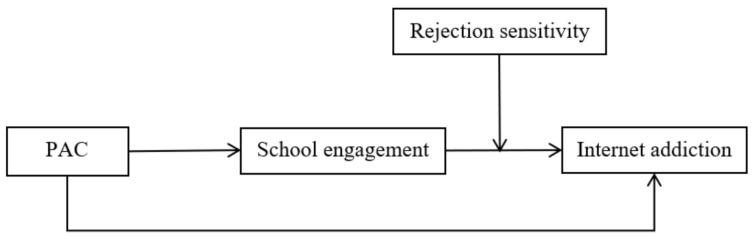
The proposed mediated moderation model. *Note*: PAC = parent-adolescent communication.

**Figure 2 ijerph-18-03542-f002:**
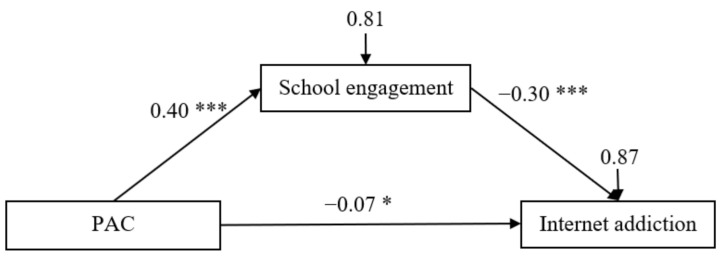
Model of the mediating role of school engagement between parent-adolescent communication and Internet addiction. *Note*: PAC = parent-adolescent communication. Values are standardized coefficients. Paths between gender, age, sensation-seeking, and each of the variables in the model are not displayed. Of those paths, the following were significant: effect of sensation-seeking on school engagement (*β* = −0.13, *t* = −4.29, *p* < 0.001, 95% CI [−0.19, −0.07]) and Internet addiction (*β* = 0.10, *t* = 2.97, *p* < 0.01, 95% CI [0.03, 0.16]). * *p* < 0.05, *** *p* < 0.001.

**Figure 3 ijerph-18-03542-f003:**
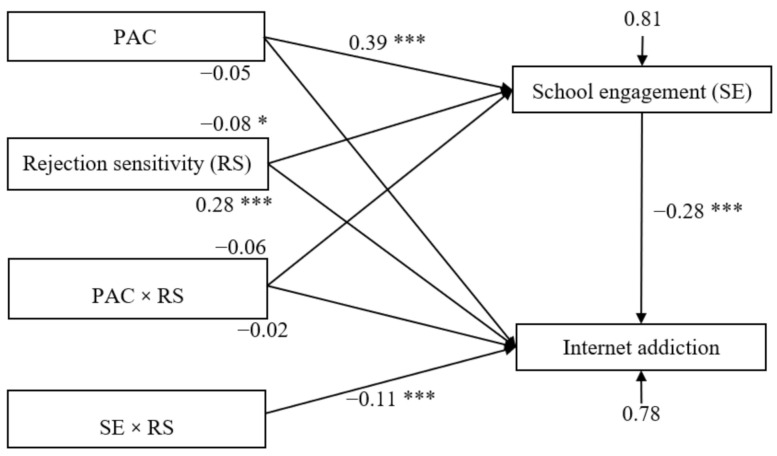
Model of the moderating role of rejection sensitivity in the indirect relationship between parent-adolescent communication and Internet addiction. *Note*: PAC = parent-adolescent communication. Values are standardized coefficients. Paths between gender, age, sensation-seeking, and each of the variables in the model are not displayed. Of those paths, the following were significant: the effects of sensation-seeking on school engagement (*β* = −0.12, *t* = −4.04, *p* < 0.001, 95% CI [−0.18, −0.06]) and Internet addiction (*β* = 0.08, *t* = 2.53, *p* < 0.05, 95% CI [0.02, 0.14]); gender on Internet addiction (*β* = 0.12, *t* = 4.17, *p* < 0.001, 95% CI [0.07, 0.18]). * *p* < 0.05, *** *p* < 0.001.

**Figure 4 ijerph-18-03542-f004:**
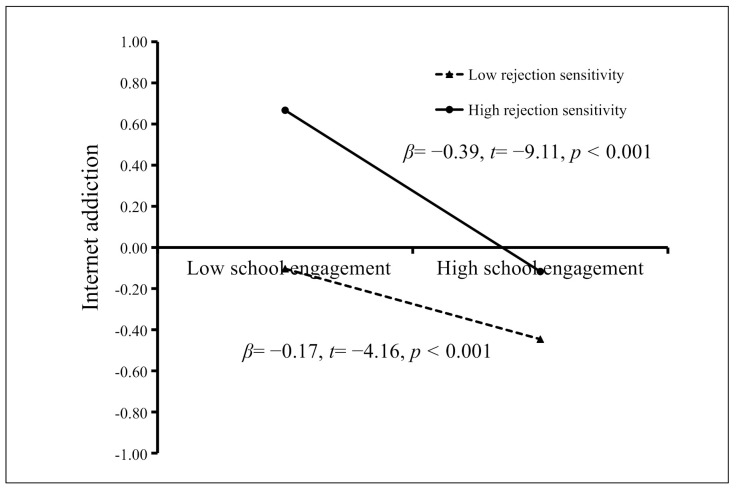
Internet addiction among adolescents as a function of school engagement and rejection sensitivity.

**Table 1 ijerph-18-03542-t001:** Descriptive statistics and correlations for all variables.

Variables	1	2	3	4
1. PAC	1.00			
2. Rejection sensitivity	−0.13 ***	1.00		
3. School engagement	0.41 ***	−0.14 ***	1.00	
4. Internet addiction	−0.20 ***	0.29 ***	−0.34 ***	1.00
*Mean*	2.26	3.04	3.91	1.26
*SD*	0.53	0.43	0.49	0.28

*Note*: PAC = parent-adolescent communication. *** *p* < 0.001.

## Data Availability

The data presented in this study are available on request from the corresponding author.

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
