# Peer review of "Parent-Adolescent Communication, School Engagement, and Internet Addiction among Chinese Adolescents: The Moderating Effect of Rejection Sensitivity"

_ijerph, 2021, doi:10.3390/ijerph18073542_

Round 1
Reviewer 1 Report
IJERPH 1137523 peer review. 23rd of February 2021.
The study submitted by Li et al. aimed at understanding the links between parent-adolescent communication, school engagement and internet addiction in participants recruited from Chinese schools.
While the study is easy to understand and follow, several alterations are needed before publication. These are outlined below.
1/ The English needs to be thouroughly revised throughout.
2/ The ethics paragraph (including recruitment of participants and ethical approuval) needs to be joined within the same paragraph, at the begining of the Methods section.
3/ What are the inclusion and exclusion criteria ? For example, readers would assume that long period of school leave should be considered as an exclusion criteria.
4/ The Internet Gaming Addiction questionaire was applied after modifications. Can the authors please comment on 'gaming', since adolescents could be addicted to Internet while not using the internet for gaming purposes (social networks, etc...).
5/ The methodological paragraph related to data analysis and statistics should be separated from the paragraph mentionning the ethical statements.
6/ Have the authors checked for Internet use for school purposes ? One biais could arise from not quantifying Internet use for academic purposes (school work, homework, research, etc...).
7/ Lines 60-62. This sentence makes no sense. Please edit.
8/ Lines 82. "deviant" should be avoided. Please be more specific.
9/ Lines 279-281, lack of foundations. Are these the results from the current study ? If not, references should be added.
10/ As a conclusion, the authors could use a Figure (similar to Figure 1) to depict their findings (with a colour-code for negative/positive correlations).
Author Response
Response to Reviewer’s Comments
The study submitted by Li et al. aimed at understanding the links between parent-adolescent communication, school engagement and internet addiction in participants recruited from Chinese schools.
While the study is easy to understand and follow, several alterations are needed before publication. These are outlined below.
1/ The English needs to be thoroughly revised throughout.
Response: Thank you very much for your advice. Your advice have contributed a lot to our article. We invited professionals to polish the language of the manuscript. The edit was performed by professional editors at Editage, a division of Cactus Communications.
2/ The ethics paragraph (including recruitment of participants and ethical approval) needs to be joined within the same paragraph, at the beginning of the Methods section.
Response: Thank you very much for your advice. Your advice have contributed a lot to our article. We have added ethics paragraph at the beginning of the Methods section. New statement in manuscript is: “The survey materials and study procedures were approved by the Ethics in Human Research Committee of the Department of Psychology, Guangzhou University (protocol number: GZHU2019012; date of approval: 2019/05/27). ”
3/ What are the inclusion and exclusion criteria ? For example, readers would assume that long period of school leave should be considered as an exclusion criteria.
Response: Thank you very much for your advice. Your advice have contributed a lot to our article. Before we carried the study, we excluded students who had been leaved school for long periods, and who were mentally ill etc. But in the actual investigation,we have not met situation like this.
4/ The Internet Gaming Addiction questionnaire was applied after modifications. Can the authors please comment on 'gaming', since adolescents could be addicted to Internet while not using the internet for gaming purposes (social networks, etc...).
Response: Thank you very much for your advice. Your advice have contributed a lot to our article. The aim of this instrument is to assess the severity of Internet Gaming Addiction and its detrimental effects by examining both online and/or offline gaming activities occurring over a 12-month period. The nine questions comprising the IGDS9-SF are answered using a 3-point Likert. The scores are obtained by summing the gamer’s answers with higher scores being indicative of higher degrees of gaming disorder. It is also worth noting that the main purpose of this instrument is not to diagnose IGD but to assess its severity and accompanying detri-mental effects to the gamer’s life.
Reference:
Pontes, H. M.; Griffiths, M. D. Measuring DSM-5 internet gaming disorder: development and validation of a short psychometric scale. Comput. Hum. Behav. 2015, 45, 137–143.
5/ The methodological paragraph related to data analysis and statistics should be separated from the paragraph mentioning the ethical statements.
Response: Thank you very much for your advice. Your advice have contributed a lot to our article. The methodological paragraph related to data analysis and statistics have separated from the paragraph mentioning the ethical statements. And we added institutional review board statement: “The study was conducted according to the guidelines of the Declaration of Helsinki, and approved by the Ethics in Human Research Committee of the Department of Psychology, Guangzhou University (protocol code: GZHU2019012, and date of approval: 2019/05/27).”
6/ Have the authors checked for Internet use for school purposes ? One biais could arise from not quantifying Internet use for academic purposes (school work, homework, research, etc...).
Response: Thank you very much for your advice. Your advice have contributed a lot to our article. The aim of this instrument is to assess the severity of Internet Gaming Addiction and its detrimental effects by examining both online and/or offline gaming activities occurring over a 12-month period. The nine questions comprising the IGDS9-SF are answered using a 3-point Likert. The scores are obtained by summing the gamer’s answers with higher scores being indicative of higher degrees of gaming disorder. It is also worth noting that the main purpose of this instrument is not to diagnose IGD but to assess its severity and accompanying detri-mental effects to the gamer’s life.
Reference:
Pontes, H. M.; Griffiths, M. D. Measuring DSM-5 internet gaming disorder: development and validation of a short psychometric scale. Comput. Hum. Behav. 2015, 45, 137–143.
7/ Lines 60-62. This sentence makes no sense. Please edit.
Response: Thank you very much for your advice. Your advice have contributed a lot to our article. We changed the statement. New statement is “Empirical evidence showed that poor parent-adolescent communication positively predicted adolescent Internet addiction [21,22] because adolescence who lack family warmth will meet their psychological needs in other ways and Internet is the most convenient way for adolescents to meet their psychological needs [23]. ”
Reference:
[21] Kang, J.; Park, H.; Park, T.; Park, J. Path Analysis for Attachment, Internet Addiction, and Interpersonal Competence of College Students. Com. Comput. Info. Sci. 2012, 342, 217-224.
[22] Soh, P. C.H.; Chew, K. W.; Koay, K. Y.; Ang, P. H.Parents vs peers' influence on teenagers' Internet addiction and risky online activities. Telemat. Inform. 2018, 35, 225-236.
[23] Wang, M. T.; Fredricks, J. A. The Reciprocal Links Between School Engagement, Youth Problem Behaviors, and School Dropout During Adolescence. Child. Dev. 2014, 85, 722-737.
8/ Lines 82. "deviant" should be avoided. Please be more specific.
Response: Thank you very much for your advice. Your advice have contributed a lot to our article. We changed the statement. New statement is “This is because school engagement limits the energy and time available for adverse activities (such as,smoking, alcohol abuse etc. ) and strengthens individuals’ connections to their institutions [23,30]. ”
Reference:
[23]Wang, M. T.; Fredricks, J. A. The Reciprocal Links Between School Engagement, Youth Problem Behaviors, and School Dropout During Adolescence. Child. Dev. 2014, 85, 722-737.
[30]Ta, B.Relationship between internet addiction, gaming addiction and school engagement among adolescents. Universal. J. Educ. Res. 2017, 5, 2304-2311.
9/ Lines 279-281, lack of foundations. Are these the results from the current study ? If not, references should be added.
Response: Thank you very much for your advice. Your advice have contributed a lot to our article. We changed the statement. New statement is “The present finding enhances our understanding of how parent-adolescent communication relates to adolescent Internet addiction and contributes to identify underlying mechanism for adolescent Internet addiction. ”
10/ As a conclusion, the authors could use a Figure (similar to Figure 1) to depict their findings (with a colour-code for negative/positive correlations).
Response: Thank you very much for your advice. The conclusion of our article is :“School engagement mediates the association between parent-adolescent communication and adolescent Internet addiction. Rejection sensitivity moderates the indirect relationship between parent-adolescent communication and adolescent Internet addiction via school engagement. Specifically, this indirect effect would become stronger for adolescence with high rejection sensitivity and weaker among adolescents with low rejection sensitivity” , such as figure1 illustrated. Thank you for your suggestion. In fact, our conclusion is the content of Figure 1, which has been marked in the article. Figure 1 is our hypothesis, and the result fits our hypothesis。

Reviewer 2 Report
Dear authors! I find your study very useful. I may only suggest to use diverse methods instead on only using of interviews. Some ways for positive solution and positive engagement for school and parents relations are necessary. Ti's true that isolated consideration of the process is very reductional and not useful at all
Author Response
Response to Reviewer’s Comments
Dear authors! I find your study very useful. I may only suggest to use diverse methods instead on only using of interviews. Some ways for positive solution and positive engagement for school and parents relations are necessary. Ti's true that isolated consideration of the process is very reductional and not useful at all.
Response: Thank you very much for your advice. Your advice have contributed a lot to our article. Future studies will use diverse methods in exploring adolescent internet addiction.

Reviewer 3 Report
Review: Parent-adolescent communication, school engagement, and Internet addiction among Chinese adolescents: The moderating effect of rejection sensitivity
This is study is very important and addressing a new and concerning phenomena. The study represents a large sample of adolescents and highlights an important risk factor.
References
- It is interesting to note the very first source dated 1998 regarding internet addiction.
- Citation complies with the journal style.
- A good mix of current research and sources are used.
Strengths of the paper
- The study highlights the benefits of the research, ie, internet addiction intervention programmes.
- The study focused strongly on the educational system where students spend most of their time, linking it to parenting styles.
- Methodology is clearly delineated, outlining participant sourcing, measures used as well as the analysis.
- Ethical consideration is well attended to.
- Result section is brilliantly set out.
- There is sufficient use of Tables and Figures and it always adds to the value of a manuscript.
- The Discussion section is a summary of the results and findings.
Weakness of the paper
- The paper starts poorly in the introduction. Perhaps a pre-preamble or a background to the study will clarify.
Major Points
- Expand on the Introduction, by perhaps including a “Background to the study”.
Minor Points
- Line 29 – 30 seem incomplete and is difficult to understand.
- Some language editing required, ie, line 1, 319.
Author Response
Response to Reviewer’s Comments
Weakness of the paper
The paper starts poorly in the introduction. Perhaps a pre-preamble or a background to the study will clarify.
Major Points
Expand on the Introduction, by perhaps including a “Background to the study”.
Response: Thank you very much for your advice. Your advice have contributed a lot to our article. We invited professionals to polish the language of the manuscript. The edit was performed by professional editors at Editage, a division of Cactus Communications.
We expand on the Introduction. The introduction part in manuscript is following:
With the rapid development of the network, internet has become an important platform for exchanging ideas, leisure, acquiring knowledge, and entertainment. However, uncontrolled and excessive Internet use causes “Internet addiction,” which is widely referred to a person is unable to control the Internet use [1]. According to survey research, approximately 8% of adolescents in China suffer from Internet addiction [2,3,4]. Internet addiction is related to various premorbid symptoms and clinical disorders [4], which seriously threaten academic performance, sleep quality, mental health, and emotional adaptation. Furthermore, it is becoming the main global public health issue that has attracted the attention of researchers from numerous fields [5,6,7]. Therefore, to establish an effective measure, it’s necessary to identify risky factor or underlying mechanism that may make individuals more prone to Internet addiction.
Parent and family related factors inherent in it play a crucial role in prevalence Internet addiction [4]. Parent-adolescent communication(PAC) is an important aspect of family system and thereby, is closely related to Internet addiction. Based on family system theory [19], parent-adolescent communication is beneficial for individual's development because positive parent-child relationship make people think he is loved and cared, which fulfill individual 's needs for belonging and affection [4]. Individuals with a positive Parent-adolescent communication are less likely to be addicted to the Internet compared those who have a negative parent-adolescent communication with their parents [8]. According to this perspective, many studies have showed that a negative parent-adolescent communication will increase the possibility to use Internet and eventually result in Internet addiction [9] .
Recently, the mediating and moderating mechanism underlying the relation between Parent-adolescent communication and Internet addiction have begun to examine [5,9], for example, examine, deviant peer affiliation [26] and self-esteem [5]were proved as potential mediators. According to organism-environment interaction model [41], individual and environment constitute a complex system, all various factors in system do not act independently, but rely on each other and work together. School is the most important place for student development. School engagement may function as a protective factor that prevents Internet addiction and promotes healthy life. Researchers found that individuals who are attached to school, behaviorally engaged, or participate in school activities are less to be addicted to the Internet [3,4].
Therefore, based on organism-environment interaction model and family system theory, the study was designed to test whether school engagement mediated the relationship between parent-adolescent communication and adolescent Internet addiction and whether this mediating effect was moderated by sensation seeking, so as to more systematically reveal the delaying mechanism of adolescent Internet addiction, and provide a basis for scientific prevention and effective control of adolescent Internet addiction.
Reference:
1 Young, K. S. Internet addiction: The emergence of a new clinical disorder.
Cyberpsychol. Behav. 1998, 237–244.
2 Wang, Y.; Zhao, Y. ; Liu, L.; Chen, Y. ; Jin, Y. The current situation of internet addiction and its impact on sleep quality and self-injury behavior in chinese medical students. Psychiat. Invest. 2020, 17, 385-385.
3 Song, J.; Ma, C.; Gu, C.; Zuo, B. What Matters Most to the Left-Behind Children's Life Satisfaction and School Engagement: Parent or Grandparent? J. Child. Family Stu. 2018, 27, 2481-2490.
4 Spada, M. M. An overview of problematic internet use. Addic.Behav. 2014, 39, 3-6.
5 Park, S.; Kang, M.; Kim, E. Social relationship on problematic Internet use (PIU) among adolescents in South Korea: A moderated mediation model of self-esteem and self-control. Comput.Hum. Behav. 2014, 38, 349-357.
6 Peris, M.; Barrera, U. D. L.; Schoeps, K.; Montoya-Castilla, I. Psychological Risk Factors that Predict Social Networking and Internet Addiction in Adolescents. Int. J. Environ. Res. Public. Heal, 2020, 17, 4598.
7 Weinstein, A.; Lejoyeux, M. Internet addiction or excessive internet use. Am. J. Drug. Alcohol. Ab. 2010, 36, 277-283.
8 Venkatesh, V.; Sykes, T. A.; Chan, F. K. Y.; Thong, J. Y. L.; Hu, J. H. Children's internet addiction, family-to-work conflict, and job outcomes. Mis. Quart. 2019, 43, 903-927.
9 Xu, J.; Shen, L. X.; Yan, C. H.; Hu, H.; Yang, F.; Wang, L.; Liao, X. P. Parent-adolescent interaction and risk of adolescent internet addiction: a population-based study in Shanghai. Bmc Psychiatry, 2014, 14, 112.
19 Bowen, M. The use of family theory in clinical practice. Comprehensive. Psychiatry. 1966, 7, 345-374.
26 Tian, Y.; Yu, C.; Lin, S.; Lu, J.; Liu, Y.; Zhang, W. Sensation Seeking, Deviant Peer Affiliation, and Internet Gaming Addiction Among Chinese Adolescents: The Moderating Effect of Parental Knowledge. 2019, Front. Psychol. 9.
41 Cummings, E. M., Davies,P. T. ,& Campbell, S. B. Developmental psychopathology and family process: Theory, research, and clinical implications. 2002, New York: The Guilford Press.
Line 29 – 30 seem incomplete and is difficult to understand.
Response: Thank you very much for your advice. Your advice have contributed a lot to our article. We invited professionals to polish the language of the manuscript. The edit was performed by professional editors at Editage, a division of Cactus Communications. We changed the statement. New statement is in line 29-30 is: “With the rapid development of the network, internet has become an important platform for exchanging ideas, leisure, acquiring knowledge, and entertainment. ”
Some language editing required, ie, line 1, 319.
Response: Thank you very much for your advice. Your advice have contributed a lot to our article. We invited professionals to polish the language of the manuscript. The edit was performed by professional editors at Editage, a division of Cactus Communications. We changed the statement. New statement is in line 319 is: “This reminded researchers to address different factor roles influencing internet addiction simultaneously. ”
